# Hydroponic Agriculture and Microbial Safety of Vegetables: Promises, Challenges, and Solutions

Shlomo Sela Saldinger [1],*, Victor Rodov [1],* , David Kenigsbuch [1] and Asher Bar-Tal [2]

[1] Institute of Postharvest and Food Sciences, The Volcani Institute, Agricultural Research Organization (ARO), P.O. Box 15159, Rishon LeZion 7505101, Israel

[2] Institute of Soil, Water and Environmental Sciences, The Volcani Institute, Agricultural Research Organization (ARO), P.O. Box 15159, Rishon LeZion 7505101, Israel

\* Correspondence: shlomos@volcani.agri.gov.il or s.sela@mail.huji.ac.il (S.S.S.); vrodov@volcani.agri.gov.il (V.R.)

**Abstract:** Hydroponics is a farming technique for growing plants with mineral nutrients using a soil-free medium. The plant roots are submerged in soil-free media, such as vermiculite or perlite, or just in mineral nutrient solutions. This allows for high production yields throughout the year with less water and agro-chemical inputs. Consequently, hydroponics is considered a sustainable agriculture technology. Hydroponically grown crops are usually protected from the diseases transmitted through soil or animals in open fields. Therefore, they require fewer chemicals for pest control and are safer than conventionally grown crops in terms of possible chemical contamination. Nevertheless, hydroponics guarantees neither plant health nor the microbial safety of fresh produce. In the case of microbial contamination by human pathogens, unlike soil-grown crops, the pathogens may rapidly spread through the circulating water and simultaneously infect all the plants in the facility. This review summarizes the up-to-date knowledge regarding the microbial safety of hydroponically grown crops and discusses the role of the hydroponic system in reducing the microbial hazards for leafy and fruity crops as well as the potential risks for contamination by human pathogens. Finally, it outlines the approaches and the available science-based practices to ensure produce safety. The contamination risk in hydroponic systems may be diminished by using novel planting materials and the appropriate decontamination treatment of a recirculating liquid substrate; by modulating the microbiota interactions; and by following strict phytosanitary measures and workers' hygienic practices. There is a timely need to adopt measures, such as the Good Agricultural Practice (GAP) guidelines, to mitigate the risks and ensure safe hydroponically grown vegetables for consumers.

**Keywords:** hydroponics; safety; microbial; fresh produce; vegetables; soilless; indoor farming; food safety; risk; human pathogens; plant

## 1. Introduction

In recent decades, fruits and vegetables have been increasingly involved in outbreaks of foodborne pathogens [1–3]. The contamination of fresh produce with foodborne pathogens, such as *Salmonella*, toxigenic *Escherichia coli*, and *Listeria monocytogenes*, can result in foodborne illness, often life-threatening, especially in immunocompromised and susceptible populations [4,5]. Foodborne outbreaks related to fresh produce cause substantial economic damage due to loss of working days, medical costs, product recall, cleaning and disinfection of the packing facility, and long-term reduction in sales due to continued concern regarding the safety of fresh produce [6]. Crops consumed raw, e.g., leafy greens, are more likely to transfer foodborne infections than vegetables that undergo cooking [7]. This review is an attempt to provide a timely overview regarding the potential microbial risks associated with the developing field of hydroponics and to discuss possible measures that may be used for ensuring the safety of hydroponically grown vegetables. It is focused primarily on recent research and is not meant to be a systematic literature review.

As microbial risks have been comprehensively studied in conventional agriculture, we begin with a brief overview regarding the microbial safety hazards in conventional farming and then present the hydroponics technologies and discuss the potential microbial risks and their sources. Finally, we discuss strategies to mitigate vegetable contamination, and we discuss the future perspectives.

**2. Microbial Safety Hazards in Conventional Farming**

In order to mitigate produce contamination, it is vital to know the potential sources of human pathogens. Produce contamination may occur across the production chain from the farm through to the processing, distribution, and the consumer's table. Frequently, the produce becomes contaminated at the preharvest stage in the field or during postharvest processing [8–10]. There are plenty of sources of contamination with foodborne pathogens on the farm [10–13]. These include contaminated water, soil, and soil amendments, such as untreated or partially treated manure fertilizer, the proximity of poultry and livestock facilities, and contact with wildlife [7,13]. Above all, worker hygiene and education are key factors for assuring produce safety at the farm level [10,13–15]. Contaminated irrigation water is considered the most important risk factor for preharvest produce contamination [7,13,16].

Groundwater and surface water exposed to external environmental contamination and used for crop irrigation may act as a vehicle for transmitting waterborne human pathogens to agricultural produce. The pathogens can be transferred by direct contact with the edible portion of fruits and vegetables or through soil contamination, reaching the rhizosphere. The risk of crop contamination in field cultivation depends upon the water quality (e.g., microbial population and organic load), irrigation method, crop type, and water source [7,10,13]. Overhead irrigation is considered the riskiest because the direct contact between the water and the edible portion of the crop leads to bacterial retention and colonization on the plant surface. In contrast, furrow or subsurface drip irrigation poses a lower risk of contamination [14,16]. The presence of a barrier (soil) in subsurface drip irrigation provides a hurdle for the direct contamination of the edible parts of the plant with human pathogenic bacteria, such as *E. coli* and *Salmonella* [17]. Several reports have indicated the possible transfer of human pathogenic bacteria from contaminated soil or water into the root system and subsequently into the edible aerial portion of the plant [18–20]. To date, this phenomenon has been observed only in small-scale laboratory and greenhouse experiments. Its occurrence in the field remains to be studied.

In order to mitigate fresh produce-related outbreaks, the U.S. Food and Drug Administration (FDA) has published a free and voluntary guide for farmers and processors of fresh produce on minimizing the microbial food safety hazards for fresh fruits and vegetables [21]. This guidance utilizes principles of good hygiene practices in the production of fresh fruits and vegetables and serves as the basis for Good Agricultural Practice (GAP). GAP refers to farming practices that minimize the likelihood of crop contamination in the field and throughout the harvesting and processing of vegetables and fruits [22]. GAPs include guidelines on the quality of irrigation water, the use of animal manure, contact with wildlife, and workers' health and hygiene. Adopting GAPs should therefore limit the introduction of human pathogens into the production chain. As foodborne outbreaks associated with fresh produce were still reported in the U.S. more than a decade after the publication of this guide, the U.S. government published in 2011 a new legislation termed the Food Safety Modernization Act (FSMA), which deals with several food safety areas, including produce safety. Later, the FDA released seven food safety rules under the FSMA. The Produce Safety Rule (PSR) sets the regulatory standards required for the safe growing, harvesting, packing, and storing of produce for human consumption [22]. Adoption and compliance with the rule require adequate training and education; so, this legislation's expected outcome on the safety of fresh produce will be evident only in future years.

### 3. Hydroponic Cultivation Systems and Their Advantages

Hydroponics is a farming technique for growing plants with mineral nutrients using a soil-free medium [23–25]. The plant roots are submerged in soil-free media, such as vermiculite or perlite, or just in mineral nutrient solutions [23]. In recent years, soilless plant cultivation has become a viable commercial agrotechnology for growing vegetables and ornamentals in greenhouses. Hydroponics may be divided into open and closed systems (Figure 1) [26,27]. In open systems, a nutrient solution continuously feeds the plants, and the excess solution gets drained out, and it then runs into the waste. The drained solution is collected and recirculated in closed systems to feed the plants [26,27]. One of the open systems is the static solution culture. Typically, in this system, seedlings within the plugs of the substrate are planted into the holes in a raft (such as Styrofoam) and floated on the surface of the nutrient solution (Figures 2 and 3). The solution is aerated or kept low enough for the upper parts of the roots to obtain sufficient oxygen. The nutrient solution is changed on a scheduled basis or when the concentrations of salts increase above a threshold value, as measured by electrical conductivity [28]. Another group of open systems is based on irrigation from the top of the growing medium in pots or containers [29,30]. The most common means of irrigation is a drip line, and the water flows from the top down and is drained out of the greenhouse. Applying liquid fertilizers through the irrigation system, named fertigation, is considered to be an efficient fertilization means in these systems [30,31]. In some of these systems, the drainage is collected and reused for the irrigation of another crop.

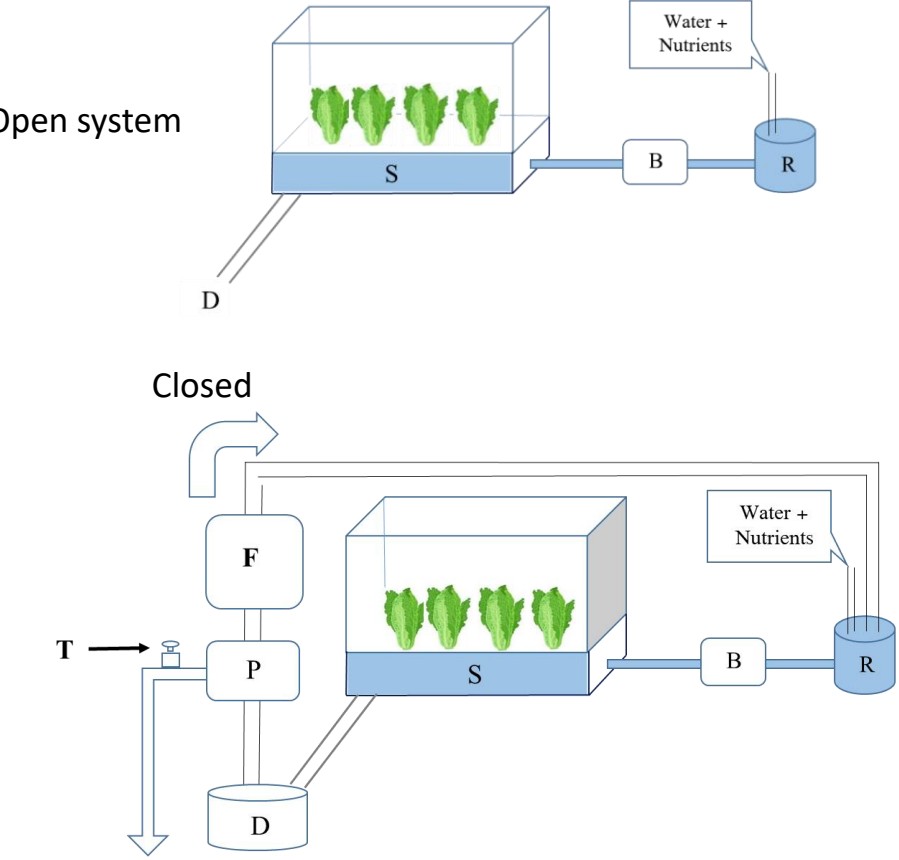

**Figure 1.** Schematic presentation of open and closed hydroponic irrigation system. R, reservoir; B, booster pump; S, substrate/solution; D, drainage; P, pump; T, tap; F, filter.

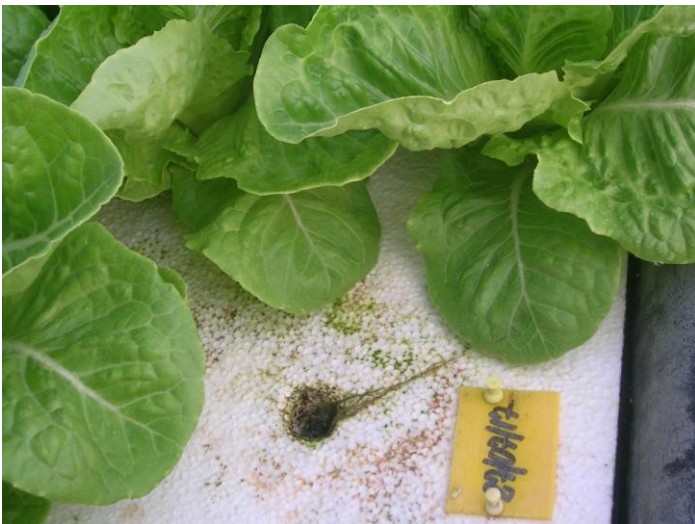

**Figure 2.** A close-up view of hydroponically grown lettuce showing the Styrofoam floating on the water and some root residues in the empty hole following harvesting of the lettuce head. Photography by E. Shalgi.

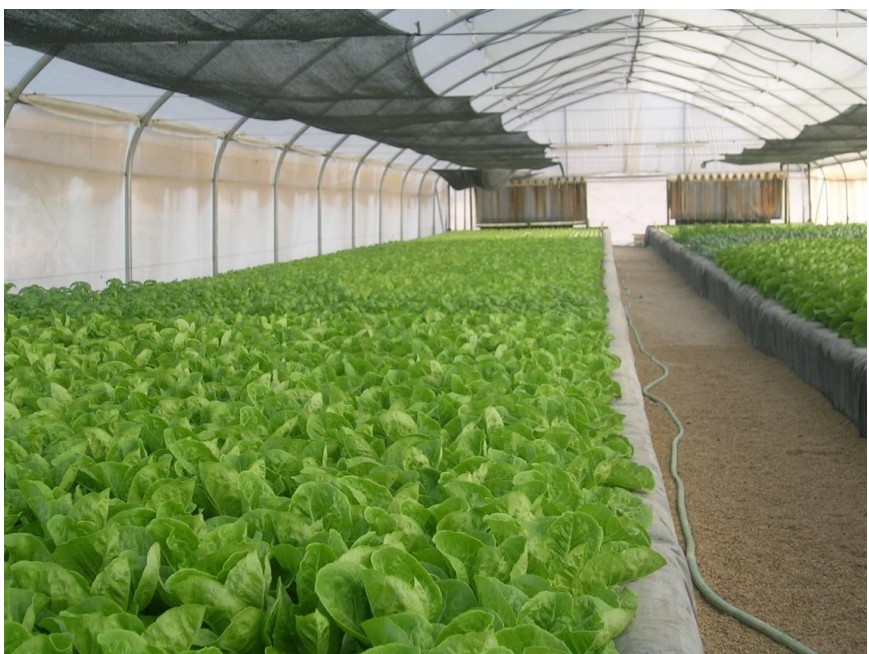

**Figure 3.** Hydroponically grown Romaine lettuce on an Israeli farm. Photography by E. Shalgi.

The closed hydroponic systems can be grouped into two main types. The first one is the continuous-flow solution culture. In this system, the nutrient solution continuously flows through the roots. The nutrient solution stock can be monitored for temperature, pH, and nutrient concentrations and adjusted automatically before feeding [24,28]. The most common system is the nutrient film technique (NFT), in which a shallow level of the nutrient solution flows in a sloped channel into a reservoir and is then recirculated [32,33]. The shallow level of the solution enables a sufficient rate of oxygen diffusion into the solution to meet the roots' consumption [24,32]. Another system is the ebb and flow (flood and drain) technique, composed of plants growing in pots in a tray filled with a soilless substrate above a reservoir of nutrient solution [32]. The tray is filled periodically with the nutrient solution, draining back into the reservoir and recirculating using a pump. In addition to these common techniques, there are other indoor farming techniques, such as aquaponics,

aeroponics, and bioponics [34]. Hydroponics offers numerous agro-technological, environmental, and economic advantages [34]. The increased world population in the next few decades and climate changes will have a profound and direct impact on agricultural and food systems. Global warming and declining precipitation in semiarid regions result in water scarcity, likely reducing yields for major crops in the coming decades. These changes could substantially impact food security [34,35]. Hydroponic agriculture substantially saves the irrigation water and fertilizers required for plant growth. Hydroponics requires 10–16% of the water necessary to produce the exact yield of vegetables by conventional agriculture. The continuous recycling of the nutrient solution in closed hydroponic systems fits well into the 'zero-waste' approach of sustainable agriculture by reducing the soil and aquifer pollution from the excess fertilizers used in open-field farming [23,26,36]. Similarly, with the lack of soil, there is no need for weeding and the usage of the herbicides that contribute to environmental pollution and the potential contamination of the food chain [23,25]. The lack of soil is also believed to reduce the plant diseases associated with soilborne pathogens, thus minimizing the need for pesticides [23,25,34].

Plants in hydroponics do not have to compete for resources. Hence, they can be planted more densely and vertically, thus saving space [34]. Hydroponic systems with controlled light and temperatures allow a shorter growth cycle compared to growth in soil, enabling several growth cycles with high and stable yields all year round [23,32]. In addition, the harvest of plants is easy to manage. Another advantage of hydroponics is the ability to grow plants in locations closer to the consumers, thereby keeping the freshness of produce and minimizing the energy and chemical usage associated with postharvest treatments, transportation, and storage. Hydroponics may also serve as urban agriculture [23,25,32]. It may provide an economical solution for growing vegetables in areas with limited land available for conventional horticulture and other farming inputs. Finally, hydroponics requires less labor-intensive practices compared with conventional agriculture.

Notably, due to the high cost of controlling light and temperature, the need for a high level of maintenance, and the mild winter climate, the majority of the Israeli and other Mediterranean region growers involved in hydroponics are currently using uncontrolled growth systems.

Altogether, hydroponic agriculture is widely accepted by the public and the authorities as a farming approach that supports sustainability, reduces crop contamination, and minimizes the environmental pollution caused by the excessive use of agrochemical amendments, such as fertilizers, pesticides, and herbicides [23,25,34]. Concurrently, the lower usage of agricultural amendments reduces potential crop contamination, thus enhancing the chemical safety of hydroponically grown plants.

## 4. Role of Hydroponic Systems in Reducing the Microbial Hazards for Vegetable Crops

### 4.1. Mechanism of Hydroponics for Reducing the Microbial Hazards

Unlike conventional farming, soilless indoor cultivation (e.g., hydroponics) offers a more controlled environment, which is easier to manage and prevents microbial contamination in the cultivation facility [37]. Growing crops in controlled-environment facilities ('plant factories') is therefore considered a safer cultivation technology than conventional farming [38]. In line with this assumption, the FDA recently exempted hydroponically and greenhouse-grown produce, labeled as 'indoor-grown', from its general recommendation to avoid the consumption of romaine lettuce from the Salinas, CA region during the 2019 outbreak of the highly virulent strain of *E. coli* O157:H7 [39]. Furthermore, consumers also perceive the produce grown under controlled 'plant factory' conditions as a safer alternative to soil-grown regular agricultural produce [40]. The lack of soil and the closed culture facility provide a significant advantage in minimizing soilborne diseases and the contamination of plants by soilborne human pathogens. Unlike soil-based culture, hydroponics does not utilize raw or partially treated (composted) animal manure, which is a known vector for the transmission of foodborne pathogens, such as *Salmonella* and pathogenic *E. coli* [41]. Indoor cultivation also protects plants from contact with soilborne

microorganisms through dust particles that may pose a contamination risk under field conditions. The proper management of the closed cultivation facility also restricts contact with insects, domestic animals, birds, and wildlife, which are known vectors of foodborne pathogens in conventional agriculture [11,13,42]. Indeed, several studies have demonstrated a lower level of contamination and the absence of generic *E. coli* and *Salmonella* in soilless-grown crops compared to soil-grown ones [43–45]. In one study, low levels of *E. coli* were detected in 3 out of 50 circulating water samples of five experimental hydroponic systems. In contrast, neither *E. coli* nor *Salmonella* was detected in 29 water samples from three experimental aquaponic cultivation systems [44]. The importance of the controlled environment in protecting hydroponically grown crops against external contamination was demonstrated in the study by Orozco et al., who described the entry of runoff water and wild animals into several hydroponic tomato greenhouses [46].

### 4.2. Effect of Culturing Technology on Reducing the Microbial Hazards

Irrigation water is a well-known source of plant contamination by human pathogens in conventional agriculture [7,13,16,47,48]. Still, several studies reported that the edible parts of vegetables grown hydroponically were free of human pathogens, even in cases where contaminated water was used for cultivation [44,49–51]. Selma and co-workers found that the microbial quality of several lettuce cultivars grown hydroponically was better than that of the soil-grown lettuce [52]. A recent literature review found no reports of foodborne pathogen contamination in aquaponics products [53], even though the water source in the aquaponics systems was derived from fish farming. Human pathogens can contaminate plants by several mechanisms. These include external root colonization, root internalization, and transmission of the pathogen to the edible aerial parts of the plants, as well as external contamination of the edible parts by overhead irrigation or accidental contact with the irrigation water [13]. Hydroponic production methods based on NFT and deep-water culture with floating rafts inherently prevent contact between the water and the aerial parts of the plants, which exerts a similar effect to that of subsurface drip irrigation systems in conventional farming [48], thereby reducing the potential risk associated with the direct contact of leaves or fruit with contaminated irrigation water. Lopez-Galvez and co-workers analyzed the microbial quality of several water sources used in hydroponic production using soilless substrate (coconut fiber and rock wool) and detected *E. coli*, presumptive *Salmonella* spp., and *Listeria monocytogenes*; however, no pathogens were found on the tomatoes despite the presence of generic *E. coli* and presumptive *Salmonella* in the water [49]. A recent study found a higher microbial load in harvested lettuce leaf samples compared to that in preharvest samples in an NFT-based system. The researchers indicated that this finding was probably due to the accidental transfer of bacteria from the contaminated root ball to the leaves during harvest and/or packaging [54]. This report highlights the role of the hydroponic culture system in preventing the cross-contamination of the edible parts of leafy greens during the growth period and calls for more caution during harvest and postharvest handling.

### 4.3. Effects of Nutrients Solutions Circulation on Microbial Hazards under Hydroponic Systems

Closed hydroponic systems involve water and nutrient solution circulation, which enables significant savings in water and nutrient usage and provides optimal nutrient concentration to each plant in the facility [34]. The closed system utilized relatively low water quantities for plant cultivation, enabling strict and better water quality control. Continuous water treatment minimizes the risk associated with transient contamination often occurring in surface water sources, which are affected by environmental and climatological factors [16]. Furthermore, the lower water needs of closed compared to open hydroponic systems and conventional farming requires a smaller and potentially more economical treatment plant to ensure water safety. Still, the accumulation of organic material in circulating water systems may negatively affect the treatment efficacy, which requires treatment modifications to remove microbial pathogens efficiently.

**5. Potential Sources and Routes of Contamination in Hydroponic Cultivation Systems**

*5.1. Potential Sources of Human Pathogens in Hydroponic Production Systems*

Although hydroponic cultivation systems are generally assumed to be safer than open field culture, as discussed above, recent recalls and outbreak investigations pointed at hydroponically grown leafy greens as the culprit [55–57]. An investigation report on the factors potentially contributing to the contamination of leafy greens grown in deep culture on reusable polystyrene rafts that were implicated in an outbreak of *Salmonella* Typhimurium during the summer of 2021 pointed at several possible sources and routes of contamination [55]. These included the presence of the outbreak strain in two nearby stormwater retention ponds located outside the growing facility as well as the inadequate storage of nutrient-rich growth media, the presence of *Salmonella* (not related to the outbreak strain) in the growing pond water, the lack of a procedure or systematic approach to ensure adequate pond water treatment, and an inadequate design and maintenance of the operation. However, the specific source and routes of leafy greens contamination were not identified. These findings support the notion that environmental sources of contamination in hydroponic production systems are essentially similar to those discussed above for conventional agriculture.

A wealth of research has been conducted to identify and understand the potential sources of the microbial contamination of fresh produce in conventional farming [13,42]. Controlled-environment agriculture utilizes closed production systems that physically eliminate the introduction of multiple external sources of crop contamination, e.g., soil or raw or partially treated manure, flood water, dust, and contact with domestic and wild animals. However, if not adequately maintained, stormwater and wild animals can enter the facility, posing a microbial risk [46]. These findings reinforce the need for appropriate construction and maintenance measures to prevent the penetration of exogenous biotic and abiotic contamination factors into the greenhouse area. It should be noted, however, that in some cases, hydroponic systems are also located outdoors, e.g., under a shelter. Such systems are inherently more prone to environmental contamination, in a similar way to open-field cultivation.

A literature search revealed a limited number of studies on the potential sources of contamination in commercial hydroponic cultivation systems (Table 1). In addition to accidental exposure to the external environment, as discussed above [46,58], other sources of contamination in hydroponic production systems were the water or nutrient solution [54,59–62], the cultivation matrix [63,64], and the poor hygiene of the workers [59].

Water is the basis of hydroponic technology. Therefore, water-associated contamination routes deserve special consideration. Hydroponics often uses high-quality municipal or desalinated tap water, which poses a low risk of transmitting human pathogens to crops. However, the utilization of untreated groundwater, surface water, and partially treated reclaimed wastewater may introduce a health risk, as these types of water often harbor microbial pathogens [7,47,48,65]. Indeed, several studies have demonstrated the presence of human pathogens in the water systems of commercial hydroponic systems [49,54,58,60,61]. In one study, a microbiological analysis was performed in hydroponic greenhouses growing bell peppers in Mexico during three production seasons in 2009–2010. The tested materials included the coconut fiber, knives, conveyor belts, pepper transportation wagons, air, water, nutrient solution for plant irrigation, and bell pepper fruits. *E. coli* was detected in 27 ($n$ = 528) of the pepper samples. *Salmonella* was isolated from only one sample ($n$ = 161) of the conveyor belt and four pepper samples ($n$ = 132). *Listeria monocytogenes* were not detected in any sample [59]. The source of the contamination was unknown and could be associated with the use of the contaminated nutrient solution or with the poor sanitary practice of the workers during harvesting and/or packaging. Another investigation in Mexico examined the microbiological condition of a hydroponic tomato farm comprising 14 greenhouses and a packinghouse implementing a high technological level and sanitary agricultural practices. The study found that 2.8% of the tomatoes were contaminated with *Salmonella* and 0.7% with *E. coli* [58]. The analysis revealed that contamination occurred

during runoff water entry into the structures and the occasional entry of wild animals into some of the greenhouses. *Salmonella* and *E. coli* were detected in tomatoes, water puddles, soil, shoes, and the feces of local wild and farm animals [46].

In another study, the sources of lettuce contamination in a closed hydroponic system were investigated by Dankwa et al. [54]. All the samples were negative for *Listeria* spp. Nevertheless, similarly to Weller et al. [44], they showed high counts of coliforms, indicating the potential presence of enteric pathogens. The investigation pointed at seedling substrate plugs as a possible source of contamination, which was further transferred to the roots and, upon harvest, to the edible leaves [54].

Contaminated seeds constitute a significant source of sprout and microgreen contamination [63,66–68]. The sprouts are grown under environmental conditions (mesophilic temperature and high humidity) that support microbial proliferation [69]. Therefore if human pathogens are introduced to the facility through seeds, they can rapidly multiply and contaminate the sprouting seeds and the entire facility. Hydroponically grown microgreens and mature leafy vegetables are also produced under environmental conditions, which may facilitate the multiplication of human pathogens. Indeed, seeds, seedlings, and seedling substrates were shown to be associated with water contamination by *L. monocytogenes* [54]. Thus, special attention should be given to the microbial safety of seeds destined for hydroponic culture.

**Table 1.** Sources and routes of contamination in commercial hydroponic farms.

| Hydroponic Systems | Crops | Microbial Hazards | Contamination Sources and Routes | References |
|---|---|---|---|---|
| Open system, pots; substrate: vermiculite | Tomato | *E. coli, Salmonella* | Pathogens were detected on tomatoes, water puddles, shoes, and local wild and farm animals; suspected sources: flood and wild animals. | [46,58] |
| Open system; substrates: coconut fiber and rock wool | Tomato | *E. coli*, *Salmonella* spp., *Listeria* spp. | *E. coli* was present in higher levels in reclaimed and surface water. Presumptive *Salmonella* spp. were detected in 7.7% of the water samples, mostly from reclaimed water. *Listeria* spp. numbers increased after adding the fertilizers. No pathogen detected on tomatoes. | [49] |
| Open system; substrate: coconut fiber | Bell pepper | *E. coli* | *E. coli* was present in higher levels in reclaimed and surface water. No link between *E. coli* prevalence and levels in water and pepper contamination. *E. coli* was present in fertilizer solutions and in water sprayed in humidifiers. | [60] |
| Open system, water from local wells; substrate: rockwool blocks, trickle irrigation | Cucumber | Fecal indicators: *E. coli*, total and fecal coliforms, *Clostridium perfringens* | *E. coli* and fecal coliforms were present on roots but only once detected on fruit. Suspected source: well water. | [62] |
| Deep culture. Not defined | Leafy greens | *Salmonella* Typhimurium | *S.* Typhimurium was isolated form 31 patients and linked to hydroponically grown leafy greens. The outbreak strain was detected in two nearby stormwater retention ponds. | [55] |
| Unknown; purchased from retail stores | Lettuce | *E. coli* O157:H7, *Salmonella*, *L. monocytogenes* | The three pathogens were detected in a number of lettuce samples. Source unknown. | [70] |
| Lettuce samples were obtained from retailers | Lettuce | Total count, coliforms, *E. coli*, yeast, mold | Aquaponically grown lettuce had significantly lower concentration of spoilage and fecal microorganisms compared to in-soil-grown lettuce. | [43] |

**Table 1.** *Cont.*

| Hydroponic Systems | Crops | Microbial Hazards | Contamination Sources and Routes | References |
|---|---|---|---|---|
| Closed system, NFT; substrate: peat moss | lettuce | Indicator bacteria, *Listeria* spp. | Substrate, roots, and seedling water reservoir harbored high counts. No *Listeria* spp. was detected. Postharvest contamination of leaves occurred, potentially due to the transfer from substrate. | [54] |
| Closed system; combination of NFT and deep water culture compared to soil-based farm | Lettuce | *Salmonella, E. coli, Stenotrophomonas maltophilia* | *E. coli* and *Salmonella* detected in 7 and 4 (out of 50) water samples, respectively. All lettuce samples (25) had <10 CFU *E. coli*/g. One lettuce sample harbored *Salmonella*. | [61] |
| Closed system, pots; substrate: coconut fiber | Bell pepper | *E. coli, Salmonella, L. monocytogenes* | *E. coli* and *Salmonella* were detected on peppers. *Salmonella* was also present on conveyor belt; suspected source: nutrient solution; poor worker hygiene. | [59] |

*5.2. Fate and Transmission Modes of Human Pathogens in Hydroponic Production Systems*

Understanding the behavior of foodborne pathogens in hydroponic systems is a prerequisite for developing control strategies. Numerous studies have focused on understanding the complex interactions between human pathogens and plants and their fate in the field [5,13,42,71–74]. In contrast, relatively few studies have assessed the behavior of human pathogens in hydroponic systems; however, there has been an accumulation of new studies in recent years (see Table 2), which underlines the importance of this topic.

A recent review by Riggio and co-workers on the microbial risks associated with leafy vegetables grown in lab-scale hydroponic systems highlighted the hazards associated with the uptake of human pathogens from the contaminated nutrient solution through the roots into the edible parts of the plant [75]. Indeed, a microbial survey in five soil-based and five hydroponic farms in Singapore was recently conducted to obtain baseline data for the urban agriculture production of lettuce [61]. *E. coli* was found in 9 and 7 out of 25 water samples derived from conventional and hydroponic farming. *Salmonella* was isolated in 6 and 4 out of 25 water samples derived from conventional and hydroponic farming, respectively. Microbiological analysis of the lettuce grown in the two systems revealed *E. coli* in 7 out of 25 crop samples grown in soil. At the same time, no contamination was detected in the 25 crop samples produced hydroponically. *Salmonella* was found only in one sample from hydroponically grown lettuce [61]. Based on laboratory internalization experiments, the authors hypothesized that lettuce contamination by *Salmonella* was likely to occur following the translocation of the pathogen from the roots to the aerial part of the crop [61].

An increasing number of studies aim to understand the fate of human pathogens and the potential transmission routes in hydroponically grown vegetables using various experimental models, ranging from a laboratory model consisting of water in a petri dish to pilot plants consisting of NFT systems (Table 2). The plants grown hydroponically were shown to be prone to root internalization by *E. coli* and *Salmonella* [76–83], and the roots' wounding appeared to increase the internalization [78,81]. A literature review concluded that a higher internalization was observed in the hydroponically grown plants compared to those of the soil-based cultivation [84]. For example, a bioluminescence-labeled *E. coli* strain was shown to internalize spinach seedlings grown hydroponically but not in those in soil. The researchers hypothesize that these findings may be attributed to either a difference in the accessibility of the roots to internalization in the two systems or to the presence of competitive microbial communities in the soil, which inhibited internalization [79]. In a similar study, *E. coli* internalization was demonstrated in hydroponically grown spinach seedlings but not in soil-grown plants [80]. Nevertheless, this study used pasteurized soil. Thus, the involvement of the rhizosphere microbial community in the uptake of human pathogens remains questionable. In another study, using *E. coli* O157:H7-contaminated

radish seeds, the researchers demonstrated the systemic contamination of whole plants on seven-day-old seedlings (microgreens). The hydroponically grown seedlings contained a higher pathogen load than those produced on peat soil substitutes [85]. The survival of *Salmonella* Enteritidis in hydroponic nutrient solution and lettuce root internalization was associated with the nutrient solution's pH level, plant age, and pathogen load [83].

Remarkably, viruses, which cannot multiply outside the host cells, can contaminate hydroponically grown plants [86,87]. Artificial water contamination by human norovirus, murine norovirus, and Tulane virus (animal caliciviruses) resulted in viral internalization and dissemination to the shoots and leaves of lettuce [86]. Similarly, the fate of murine norovirus (human norovirus surrogate) was investigated in hydroponically grown kale and mustard microgreens [87]. Low levels of infectious virus were detected in both the edible tissues and the roots. Recirculated water maintained relatively high levels of the contagious virus throughout the harvest. It was reported that even after an initial contamination event was removed, the viruses were still present in the recirculated water; they were taken up through the roots and reached the edible tissues [87].

Taken together, these studies highlight the potential of human pathogens to persist in the water and soilless substrate and water, internalize the roots, and consequently contaminate the whole plant through systemic transmission.

Closed hydroponic systems are characterized by water circulation, which enables culture optimization while reducing water and nutrient usage [34]. Yet, water circulation may also pose a special risk factor in the case of human pathogens entering the production system. The circulating water may contain organic material from the water source (e.g., surface water, reclaimed wastewater, and fish culture), nutrient growth solution, and root secretions. The organic material coupled with mesophilic temperatures results in permissive conditions that can support the persistence and the multiplication of human pathogens upon their introduction into the water system [80,85,88–91]. As the water and the nutrient solution are continuously circulating throughout the production area, it may rapidly cause the dispersal and spreading of the pathogens in the entire facility, resulting in widespread cross-contamination of the hydroponically grown crop. The successive usage of the same water in subsequent crop cultivation will eventually result in cross-contamination of the new crop. Human pathogens can form or join existing biofilms on wet surfaces [92,93], including those of plants [11,13,33,94]. This feature may result in the establishment of pathogens in the entire facility. Even if the water is drained and replaced with fresh water, bacteria persisting within the biofilm have the potential to cross-contaminate the water of the next crop. As biofilm bacteria are more tolerant to common cleaning and disinfection measures compared to planktonic bacteria [95–97], it is critical to install maintenance programs, which should include the monitoring of the presence of human pathogens (or indicator bacteria) in the water, water treatment, cleaning, and the sanitation of surfaces in contact with water. Appropriate maintenance is a crucial factor in ensuring the safety of hydroponically grown crops.

**Table 2.** Fate of human pathogens in model and experimental systems.

| Model System | Crops | Microbial Hazards | Contamination Sources and Routes | References |
|---|---|---|---|---|
| Lab system; water in petri dish | Lettuce | *E. coli* O157:H7 strains | Water. Bacteria adhered preferentially to roots and seed coats; bacteria proliferated in seedlings. | [88] |
| Lab scale (tubes) | Lettuce | *Salmonella* Enteritidis | *Salmonella* was inoculated into the nutrient solution. The pathogen survived in the system and colonized the roots. Root internalization was higher in younger plants. pH and inoculum size affected the internalization and survival. | [83] |

**Table 2.** *Cont.*

| Model System | Crops | Microbial Hazards | Contamination Sources and Routes | References |
|---|---|---|---|---|
| Lab scale, test tube; open system | Spinach | GFP-tagged *E. coli* O157:H7 | Following inoculation of the hydroponic medium, *E. coli* was found in the roots and shoots. Concentration in shoots increased from 14 to 21 days. Internalization was observed in hydroponically grown plants but not in soil-grown plants. | [80] |
| Lab-experiments | none | *Generic E. coli* | *E. coli* strains survive but do not proliferate in irrigation water and in several fertilizer solutions. Solution containing HNO3 inactivated *E. coli*. | [60] |
| Lab system (hydroponic tray) | Tomato | *Salmonella* | Artificially contaminated nutrient solution resulted in bacterial internalization. *Salmonella* found in the hypocotyls-cotyledons, stems, and leaves of 10-day-old plants. | [77] |
| Lab scale; hydroponic trays; open system | Spinach | *E. coli* O157:H7 | Inoculation of medium resulted in root internalization and transmission to the stem and leaves. Wounding the roots increased internalization. Internalization was higher in soil- versus hydroponically grown plants. | [78] |
| Lab-scale containers; open system | Spinach | *E. coli*, *Salmonella*, *L. monocytogenes* | Two contamination routes tested; hydroponic medium and leaves. Root and leaf contamination was rare with low inoculum ($10^3$ CFU/mL); leaf, but not root, contamination was rare with high concentrations ($10^6$ CFU/leaf). Root internalization is the principal route of leaf contamination. | [82] |
| Lab experiments. (Fertilizer solution with plant was taken from a deep flow technique system) | Basil | *E. coli* O157:H7; non-O157 STEC; *Salmonella* | Inoculation of the pathogens into a fertilizer solution resulted in proliferation over 24 h. *E. coli* O157:H7 grew better in fertilizer solution with plants, while non-O157:H7 *E. coli* and *Salmonella* grew better in solutions without plants. | [90] |
| Lab-scale deep culture open system | Lettuce | *E. coli* O157:H7 | Artificially contaminated water. Internalization was observed. Root injury increased internalization. | [81] |
| Lab-scale deep culture open system | Maize (young seedlings) | *E. coli* | Artificially contaminated nutrient solution resulted in decline of counts with time. *E. coli* internalized in the roots and was detected in the shoot. | [76] |
| Experimental system; open system | Lettuce | Human norovirus, murine norovirus, Tulane virus | Artificial water contamination resulted in viral internalization and dissemination to shoots and leaves | [86] |
| Pots with peat moss substrate; growing pads; open system | Radish seedlings (microgreens) | *E. coli* O157:H7 | Artificially contaminated seeds led to systemic contamination of the seedlings in both growing systems with a higher level in the hydroponic system; they survive and proliferate significantly. | [85] |
| Micro-Mats hydroponic growing pad and mini-seed tray with overhead irrigation; open system | Swiss chard (microgreen) | *Salmonella* | Artificial contamination of seeds and irrigation water. *Salmonella* growth was affected by serovar and inoculation level; irrigation water inoculation also resulted in proliferation that was affected by initial inoculation level and the growth medium. | [98] |
| Hydroponic mats; open system | Amaranth, Broccoli, Kale, Mustard, Coriander, Rocket, Parsley, Basil, Radish | *E. coli* O157:H7 | Inoculation of seeds or water. Bacteria proliferated and colonized eight different species of microgreens. | [91] |

**Table 2.** *Cont.*

| Model System | Crops | Microbial Hazards | Contamination Sources and Routes | References |
|---|---|---|---|---|
| NFT system; closed system | Lettuce | *E. coli, Salmonella, Entamoeba histolytica, Ancylostoma* spp. | The effect of nutrient solutions on microbial quality was tested. No bacterial contamination was detected. Few samples contained *Entamoeba histolytica*, eggs, and larvae of *Ancylostoma* spp. No contamination was found when mineral nutrient solutions were used. | [99] |
| NFT system; closed system | Lettuce | S. Typhimurium L. monocytogenes | Artificial contamination of nutrient solution resulted in the persistence of the pathogens in the system throughout the growth period. The pathogens accumulated in rockwool medium and on lettuce roots and transferred to the leaves. *L. monocytogenes*, but not *Salmonella*, proliferated in the system following simulation of sporadic contamination (~104 CFU/mL) | [89] |
| NFT system; closed system | Spinach | Bioluminescence-labeled *E. coli* | Seed contamination resulted in surface and internal root colonization. The colonization was restricted to the roots in mature plants. In soil-grown plants colonization was restricted to the root surface | [100] |
| Experimental systems, water recirculation; closed system | Strawberry Basil, Lettuce | *E. coli, Salmonella* | Three out of seventy-nine water samples contained low levels of *E.coli*; no *Salmonella* was detected. | [44] |
| Experimental hydroponic closed systems | Lettuce, Basil, Tomato | Shiga-toxin-producing *E. coli, Salmonella, L. monocytogenes* | Only *E. coli* detected in water and on root surfaces. Source unknown. No contamination in edible parts. | [50] |
| Experimental; closed system | Lettuce | *Salmonella* | Seeds were artificially contaminated. *Salmonella* persisted in water and the farming environment for 6 weeks. | [101] |
| Hydroponic pads; closed system | Kale, Mustard (microgreens) | human norovirus surrogate (murine norovirus) | Inoculation of water resulted in root and leaf contamination The virus persisted in the system and caused cross-contamination. | [87] |
| Closed system, commercial | Three lettuce genotypes were studied | *Coliforms, Lactic acid bacteria* | Lactic acid bacteria and total coliform counts were lower in soilless-grown lettuce compared to soil-grown. | [52] |
| Green house. System not defined | Lettuce | *E. coli*, thermotolerant coliforms and total coliforms, *Salmonella* spp. and helminth eggs | Domestic wastewater effluents with different levels of treatment were used for irrigation. Leaves showed low levels of contamination with *E. coli*, thermotolerant coliforms, and total coliforms. *Salmonella* spp. and helminth eggs were not detected in the water. High bacterial loads were present on roots. | [51] |

## 6. Mitigation of Microbial Contamination in Hydroponically Grown Crops

Leafy and fruity vegetables are typically eaten raw and current cleaning and disinfection technologies cannot assure complete eradication of human pathogens. Understanding the potential sources and routes of microbial contamination in hydroponic facilities forms the basis for developing and implementing efficient mitigation strategies. While knowledge regarding contamination paths is still being accumulated, as discussed above, it is clear that, as with conventional agriculture, farmers must take care of all the aspects of production, processing, and packaging stages of hydroponically grown produce. As with soil culture, where guidance such as that of GAP and the FSMA legislation provides guidance to minimize the potential safety hazards in the field [102,103], specific hydroponic production practices should provide the 'know-how' regarding proper operation and management of hydroponic facilities. Recently, a new aquaponics policy was applied in the USA, and

the U.S. Department of Agriculture (USDA) has released the requirements for aquaponic operations, which are eligible to be audited under the USDA GAP audit program [104]. Many aspects of aquaponics are common with hydroponics, and it is likely that similar guidelines will be established in the near future. The application of Hazard Analysis and Critical Control Point (HACCP) in large-scale production facilities should inform production managers regarding the potential risks and the need for routine microbial monitoring. Critical control points should include, among others, all production inputs, such as irrigation water, nutrient solution, seeds, seedlings and growth substrate, and amendments (e.g., insecticides, and fungicides), as well as production surfaces and equipment used in the facility.

Finally, and perhaps most importantly, is the education and training of the workers. Understanding the potential microbial risks in the production facility, including sanitation and the workers' hygiene and the consequences of produce contamination is a key factor in the prevention of microbial hazards. A recent review concluded that continuous training is important in terms of knowledge retention and improving confidence in the implementation of food safety practices [14].

Various approaches that may be used to mitigate known microbial contamination risks in hydroponic culture are summarized in Table 3.

**Table 3.** Approaches to mitigating microbial contamination in hydroponically grown crops.

| Approach | Specific Measures | References |
|---|---|---|
| Reducing contamination of planting material | Seed decontamination: sanitizers, ozone gas, ethanol, advanced oxidation, microbubbles | [105–108] |
| | Hydrogel use as seedling growing substrate | [109–111] |
| | Antimicrobial hydrogel substrates, e.g., chitosan | [109] |
| | Natural antimicrobials added to substrates | [109] |
| | Inorganic nanoparticles added to substrates | [112–114] |
| Sanitation of recirculating water | Filtration | [47,75,115] |
| | Sanitizer additives: chlorine, iodine, $H_2O_2$, etc. | [47,75,115] |
| | Physical water treatment: UV, ultrasound | [47,75,115] |
| | Physical water treatment: plasma | [29,116] |
| Controlling plant colonization by human pathogens | Controlling plant diseases as route for human pathogen penetration: agrochemicals | [117,118] |
| | Biocontrol agents: bacteria, bacteriophages | [69,119–121] |
| | Selection of resistant plant genotypes | [122–124] |
| Controlling biofilms that harbor human pathogens | Sanitizers | [92] |
| | Surface modification | [125] |
| | Physical measures (e.g., ultrasound, UV) | [119,125] |
| | Micobiome manipulation, biocontrol | [119] |

All components of hydroponic facilities should be monitored routinely for indicator bacteria and foodborne pathogens to assess the potential risk of crop contamination. Microbial monitoring should also involve processing surfaces, instruments, and packaging materials. Hydroponic crop production using wastewater streams of low microbial quality may pose a potential safety hazard to consumers. Restricting plant growth in such settings to crops, which are not eaten raw or are not for human consumption, might be a practical alternative to prevent produce-associated diseases. Contaminated seeds are one of the major vectors introducing human pathogens into hydroponic systems [66]. Only seeds from a certified source following good agricultural practices should be used for preparing planting material for hydroponics. Furthermore, efficient seed treatment can reduce the contamination risk. Seed decontamination with a 20,000 ppm solution of calcium hypochlorite

is a standard method recommended by the FDA [107]. Various sanitizers, such as acidified sodium chlorite, quaternary ammonium, ozone gas [108], or ethanol [105], were used for seed decontamination. While none of the seed treatments resulted in immediate pathogen eradication, the plants growing from the treated seeds eventually showed less or even no contamination [108]. Promising results were demonstrated recently by advanced oxidation combining hydrogen peroxide and UV-C applications aided by microbubbles [106].

Recently, researchers have reported that the seedling substrates used for planting might carry pathogenic microorganisms and suggested that growers should use substrates that are less favorable for microbial growth [54]. For example, using polymer-based hydrogels as a substitute for traditional hydroponic substrates is an innovative practice that can tackle the food safety and sustainability challenges of soilless urban agriculture [109,110]. The hydrogel is a three-dimensional polymeric structure that can swell significantly upon water absorption and ensure moisture and nutrient retention for a long time. Teng et al. [111] presented a novel type of hydrogel with enhanced porosity for better root zone aeration, supporting clean plant growth under terrestrial and spaceflight conditions when microbiological safety is of particular importance [126]. Antimicrobial activity can be imparted to a hydrogel substrate by using a polymer base with intrinsic antimicrobial properties, such as chitosan, or by adding low-molecular bioactive materials, e.g., essential oil compounds [109]. In addition, preparing antimicrobial hydrogel substrates by incorporating inorganic nanoparticles of silver, copper, or titanium dioxide was reported by several authors [112–114]. However, the latter approach is doubtful due to the potential human and environmental toxicity of metal nanoparticles [127,128].

Appropriate water treatment is of utmost importance for ensuring the safety of hydroponic agriculture. Moreover, it may, allegedly, ensure the safe reuse of wastewater for the hydroponic production of leafy greens, e.g., lettuce [129,130]. The methods for controlling the microbial contamination of recirculated hydroponic substrate include filtration, physical treatments such as UV illumination, chlorination or the use of other antimicrobial agents (iodine, hydrogen peroxide, or ozone), and microbiome manipulation, i.e., biological control [47,75,115]. Various types of plasma treatment attract attention as potential means for water decontamination in hydroponic systems [29,116]. Each approach has advantages and drawbacks, as summarized by Riggio et al. [75]. For example, water chlorination should be carefully assessed since a previous study demonstrated that this intervention results in the accumulation of chlorates in hydroponically grown tomatoes in concentrations exceeding the EU limit [131]. Other studies raised concerns regarding the development of tolerance to disinfectants that may facilitate the emergence of antibiotic-resistant bacteria, which pose a global health concern [132].

Another safety-related aspect is the interaction of human pathogens with other microbiota in the microbial communities of hydroponic systems, particularly with plant pathogens. In contrast to popular belief, substituting soil with a liquid nutrient solution combined with that or another form of a solid substrate does not eliminate the chances of plant disease development. Moreover, it may provide preferable conditions for some species, e.g., the motile plant pathogens with 'swimming' zoospores, such as oomycete *Phytium* spp. [133]. Lesions caused by phytopathogenic agents may help human enteric pathogens such as *Salmonella* in plant colonization [12,134,135]. Therefore, plant protection measures that include, among others, agrochemical interventions may improve the microbiological safety of hydroponic systems [117,118]. In addition, plant genotypes can vary in their susceptibility to colonization by human pathogens [122,123], opening a potential for breeding cultivars with improved food safety traits [124,134].

Human pathogens can be harbored in hydroponic and irrigation facilities within biofilms composed of algal–bacterial complexes embedded in an extracellular biopolymer matrix [92]. Moreover, these biofilms can serve as a reservoir for antibiotic-resistant bacteria [33]. The capacity of several sanitizers to control the biofilm in hydroponic lettuce culture was tested by Rodriguez et al. [92] with limited success because the sanitizer doses sufficient to inhibit the biofilm manifested phytotoxicity. On the other hand, biofilm devel-

opment could be controlled by coating the surfaces with anti-fouling materials, hampering the bacterial attachment [125]. Suppression of the biofilm growth in a vertical hydroponic lettuce culture was reached by inoculating the system with a biosurfactant-producing bacterial culture *Pseudomonas chlororaphis* and by ultraviolet irradiation [119]. Biosurfactant-producing pseudomonads also showed a biocontrol activity against the major hydroponic spoilage agent *Pythium* sp. [119,120]. Furthermore, biocontrol approaches, such as the antagonist bacterium *Enterobacter asburiae* and lytic bacteriophages [121], as well as spore-forming *Bacillus* strains, proved efficient in suppressing the growth of *Salmonella* on mung bean [69,121] and alfalfa sprouts [121].

## 7. Conclusions

In recent decades, fresh produce has been increasingly involved in outbreaks due to the consumption of contaminated fruits and vegetables with foodborne pathogens. Hydroponics is a sustainable indoor agricultural technology that allows plants to grow on a soilless aqueous medium with mineral nutrients. If properly applied, it will reach high produce yields with a reduced input of agrochemicals, thus minimizing chemical crop contamination and environmental pollution. In terms of microbial food safety, it is generally considered a safer practice than conventional farming due to the lack of soil, the restricted contact with the outside environment, and the controllable 'plant factory' conditions.

Nevertheless, using hydroponics guarantees neither plant health nor food safety. The hydroponic-based technologies do not eliminate potential contamination sources such as water, seeds, seedlings, and planting substrate. Water quality is critical for the microbiological safety of hydroponic systems. The utilization of low-quality water or contaminated seedlings may introduce human pathogens into hydroponic systems; these pathogens can spread rapidly through water recirculation in closed systems and infect the whole facility much faster than in conventional soil culture. Therefore, the microbiological safety of the irrigation water, nutrient solution, seeds, and other inputs used for crop production should be carefully maintained by the appropriate treatment when needed and by routine monitoring. Strict phytosanitary measures, as well as hygienic practices and training of the employees, are critical to avoid crop contamination.

## 8. Future Perspectives

Disseminating current and future knowledge to all stakeholders involved in hydroponic farming as well as the relevant regulations should form a common infrastructure for the safe handling and growing of produce. As with the conventional agricultural technologies, there is a timely need for the developing and adopting of good hydroponics practices, which are similar to GAPs, as components of the guidelines and regulations for the safety of hydroponics-grown fresh produce. Many aspects, and especially those related to water quality, that are included in the FDA guide to minimizing microbial food safety hazards for fresh fruits and vegetables [21], as well as in the FSMA [103], are also relevant to hydroponic systems and should be adopted to ensure consumer health.

Further surveys and studies are also needed to gain baseline information on the microbial quality of hydroponic facilities and to elucidate the microbial contamination modes and routes in the various sub-types of indoor farming systems. For example, Dong and Feng [136] recently applied microbiome and bioinformatics analysis tools to evaluate food safety hazards and perform risk assessments in hydroponic and aquaponic farming systems. They have identified spoilage bacteria and potential human, plant, and fish pathogens on samples taken from fresh produce, nutrient solutions, farm tools, and farm workers. Microbiome analyses of these samples enabled them to predict the bidirectional transmission routes between plants and the surrounding environment and to construct a detailed bacteria transmission map [136]. Similar studies may be used as a basis for identifying microbial hazards toward applying the principles of the HACCP system to control the microbial safety of hydroponically grown leafy greens and fruity vegetables.

**Author Contributions:** All authors were involved in writing and editing of the manuscript. All authors have read and agreed to the published version of the manuscript.

**Funding:** This research received no external funding.

**Institutional Review Board Statement:** Not applicable.

**Informed Consent Statement:** Not applicable.

**Data Availability Statement:** No data were generated.

**Acknowledgments:** We thank Raneen Shawahna for preparing the schematic presentation of opened and closed hydroponic systems and E. Shalgi for the photographs of hydroponically grown lettuce. S. Sela Saldinger was a member of HUPLANTcontrol, COST Action 16,110 Control of Human Pathogenic Microorganisms in Plant Production Systems. He also acknowledges NIFA-BARD grant NB-8316-16 from BARD, the United States—Israel Binational Agricultural Research and Development.

**Conflicts of Interest:** The authors declare no conflict of interest.

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
