# Peer review of "Hydroponic Agriculture and Microbial Safety of Vegetables: Promises, Challenges, and Solutions"

_horticulturae, doi:10.3390/horticulturae9010051_

Round 1
Reviewer 1 Report
The manuscript is well written and provides important up-to-date knowledge regarding the main safety aspects of hydroponically grown crops. Even if is a traditionally literature review the period which was considered for the reviewed studies would be better to be specified.
Author Response
Even if is a traditionally literature review the period which was considered for the reviewed studies would be better to be specified.
- Thank you for this note. This is indeed not a systematic literature review. We have added a sentence to define the focus of the review (see, L.54-63).
Reviewer 2 Report
Dear Authors,
I enjoyed reading your manuscript and have only a few minor comments on how to improve it.
The term produce does not seem adequate, maybe a better choice would be production.
Why do chapters 5 and 6 have the same name? Please combine those two into one, or subdivide them.
Author Response
- The term produce does not seem adequate, maybe a better choice would be production.
- Produce and fresh produce are common terms in the agricultural literature referring to vegetables and fruits. Please see, for example in the cited reference list: Machado‐Moreira, Bernardino, et al. Microbial contamination of fresh produce: What, where, and how? Compr Rev Food Sci F 2019, 18(6), 1727-1750.
Even international organization handling fruit and vegetables are using this term. Please see: the international fresh produce association (IFPA), https://www.freshproduce.com
Therefore, we believe the term produce is adequate in the context of this manuscript.
- Why do chapters 5 and 6 have the same name? Please combine those two into one, or subdivide them.
- We are sorry for this technical mistake. Following the comments of Rev. 3 we have reorganized the chapters and revised the titles.
Reviewer 3 Report
Title of the paper:
The paper title does not reflect the content of this paper. However, this title is an enormous scope of the effect of hydroponics systems on microbial food safety hazards.
My suggestion to authors is to write about specific crops such as lettuce, or at least vegetables, to be more specific and to have a high contribution to the reader.
Abstract:
Lines 11 and 12: The definition of hydroponics is not correct due to hydroponics can be done indoor farming and outdoor farming and under shelter. Also, hydroponics was not growing in the soilless medium. It is growing the plants with water only. In contrast, a soilless cultivation system can be cultivated the plant under water and a substrate medium.
Lines 25 and 26: The authors mentioned a timely need to adopt Good Agricultural Practices (GAP) and Good Handling Practices (GHP) guidelines to mitigate the risks and ensure safe hydroponically grown produce for consumers. But unfortunately, the authors did not discuss the GAP and GHP role in decreasing the risks of microbial food safety hazards. In addition, how the GAP and GHP can affect the hydroponics production system.
1. Introduction
The introduction should be about the microbial safety hazards in hydroponics, not under conventional farming, or the authors can change the title to the comparison between the hydroponics system and conventional farming and microbial food safety hazards.
However, what is the reasons for the increase in the levels of microbial such as E. coli and Salmonella? From where starting this microbial infection by E. coli and Salmonella? What is the role of agronomic factors in increasing E. coli and Salmonella under conventional cultivation?
The introduction did not have the general and specific objectives of this paper.
2: The promise of indoor farming
Lines 90 until 92: Please explain that there is a low level of contamination and the absence of generic E. coli and Salmonella in soilless-grown crops compared to soil-grown. What is the role of GAP and GHP in decreasing the E. coli and Salmonella in soilless-grown crops?
3. Hydroponic cultivation systems
Lines 103 until 122: There is no need to define the hydroponics cultivation system or explain the open and closed systems. Therefore, I suggest these subtitles:
3: Role of hydroponics system in reducing the microbial hazards for leafy and fruity crops
3.1 Mechanism of hydroponics for reducing the microbial hazards
3.2 Effect of nutrient film technique and Deep water culture for reducing the microbial hazards
3.3 Effects of nutrients solutions circulation on microbial hazards under hydroponics system
4. Benefits of Hydroponics
The information in this subtitle was repeated in other parts of this paper. Therefore, I suggest preparing a table composed of the hydroponics system, the crops, the microbial hazards such as E. coli and Salmonella and the reasons for these microbial hazards as the following;
|
Hydroponic systems |
Crops |
Microbial hazards |
contamination sources and Reasons |
References |
|
Example: NFT |
Lettuce |
E .coli |
E. coli was detected in the water and on the roots |
|
7:Mitigation of microbial contamination in hydroponically grown crops
Please prepare a table and summarize the mechanism for mitigation of microbial contamination in leafy and fruity crops under hydroponics.
8:Concluding remarks and future perspectives
Please septate the concluding and future perspectives subtitles into the following:
8. Conclusion
9. Future studies
Author Response
Title of the paper:
The paper title does not reflect the content of this paper. However, this title is an enormous scope of the effect of hydroponics systems on microbial food safety hazards.
My suggestion to authors is to write about specific crops such as lettuce, or at least vegetables, to be more specific and to have a high contribution to the reader.
- Thanks for the suggestion. We have renamed the title to be more specific to vegetables.
Abstract:
Lines 11 and 12: The definition of hydroponics is not correct due to hydroponics can be done indoor farming and outdoor farming and under shelter. Also, hydroponics was not growing in the soilless medium. It is growing the plants with water only. In contrast, a soilless cultivation system can be cultivated the plant under water and a substrate medium.
- Thank you for noticing this point. We have revised the definition, as suggested (see, L12-4).
Lines 25 and 26: The authors mentioned a timely need to adopt Good Agricultural Practices (GAP) and Good Handling Practices (GHP) guidelines to mitigate the risks and ensure safe hydroponically grown produce for consumers. But unfortunately, the authors did not discuss the GAP and GHP role in decreasing the risks of microbial food safety hazards. In addition, how the GAP and GHP can affect the hydroponics production system.
- We think that the suggested discussion should be in the main review and not in the Abstract. We have extended the discussion on role of GAP in decreasing microbial safety risks in hydroponic production systems (see, L. 95-100 and 556-8).
- Introduction
The introduction should be about the microbial safety hazards in hydroponics, not under conventional farming, or the authors can change the title to the comparison between the hydroponics system and conventional farming and microbial food safety hazards.
However, what is the reasons for the increase in the levels of microbial such as E. coli and Salmonella? From where starting this microbial infection by E. coli and Salmonella? What is the role of agronomic factors in increasing E. coli and Salmonella under conventional cultivation?
The introduction did not have the general and specific objectives of this paper.
- We have revised the structure and content of the Introduction. We have added a new 'Introduction' section (new section 1), which set the scope of the review. The new section 2 discusses the state of art of our knowledge regarding the microbial safety hazards in conventional farming. We think that such knowledge may serves as a necessary background for understanding common risks and potential sources of crop contamination, before we begin to discuss microbial risks associated with hydroponics. We added a few sentences in the 'Introduction' regarding the structure of the review (see, L 54-63).
2: The promise of indoor farming
Lines 90 until 92: Please explain that there is a low level of contamination and the absence of generic E. coli and Salmonella in soilless-grown crops compared to soil-grown.
- Done (see, L224-6).
What is the role of GAP and GHP in decreasing the E. coli and Salmonella in soilless-grown crops?
- This is now discussed (see, L. 96-100; 451-64; 584-90).
- Hydroponic cultivation systems
Lines 103 until 122: There is no need to define the hydroponics cultivation system or explain the open and closed systems. Therefore, I suggest these subtitles:
3: Role of hydroponics system in reducing the microbial hazards for leafy and fruity crops
3.1 Mechanism of hydroponics for reducing the microbial hazards
3.2 Effect of nutrient film technique and Deep water culture for reducing the microbial hazards
3.3 Effects of nutrients solutions circulation on microbial hazards under hydroponics system
- Thank you for your notes and suggestions, which prompted us to re-organize the chapters that discuss microbial safety aspects of hydroponics. We added a chapter title as suggested and we believe that the present organization of the chapters increase the clarity and reading fluency.
Regarding the chapter on hydroponics systems, we think the reader will benefit from a brief description of the hydroponic systems before discussion on the role of the hydroponic systems in reducing microbial hazards.
- Benefits of Hydroponics
The information in this subtitle was repeated in other parts of this paper. Therefore, I suggest preparing a table composed of the hydroponics system, the crops, the microbial hazards such as E. coli and Salmonella and the reasons for these microbial hazards as the following;
|
Hydroponic systems |
Crops |
Microbial hazards |
contamination sources and Reasons |
References |
|
Example: NFT |
Lettuce |
E .coli |
E. coli was detected in the water and on the roots |
|
- Thank you for this suggestion. We added two new tables (Tables 1 and 2). One, which describes microbiological findings from commercial hydroponic systems, and one, which summarizes experimental lab-scale systems.
7:Mitigation of microbial contamination in hydroponically grown crops
Please prepare a table and summarize the mechanism for mitigation of microbial contamination in leafy and fruity crops under hydroponics.
-Done. See, Table 3.
8:Concluding remarks and future perspectives
Please septate the concluding and future perspectives subtitles into the following:
- Conclusion
- Future studies
- Done
Round 2
Reviewer 3 Report
The authors revised the paper according to my comments.
According to Horticultural MDPI, the references list needs to be corrected, such as ref no. [1] [10] [14] [15], and so on.
For example: [10] Gil, M.I., et al., Pre-and postharvest preventive measures and intervention strategies to control microbial food safety hazards of fresh 674 leafy vegetables. Critical Reviews in Food Science and Nutrition, 2015. 55(4): p. 453-468.
The reference must be corrected such as this reference: Gil, M. I., Selma, M. V., Suslow, T., Jacxsens, L., Uyttendaele, M., & Allende, A. Pre-and postharvest preventive measures and intervention strategies to control microbial food safety hazards of fresh leafy vegetables. Critical reviews in food science and nutrition 2015, 55(4), 453-468.https://doi.org/10.1080/10408398.2012.657808
Author Response
The references list was corrected.